# Matrix Metalloproteinases on Severe COVID-19 Lung Disease Pathogenesis: Cooperative Actions of MMP-8/MMP-2 Axis on Immune Response through HLA-G Shedding and Oxidative Stress

**DOI:** 10.3390/biom12050604

**Published:** 2022-04-19

**Authors:** Pedro V. da Silva-Neto, Valéria B. do Valle, Carlos A. Fuzo, Talita M. Fernandes, Diana M. Toro, Thais F. C. Fraga-Silva, Patrícia A. Basile, Jonatan C. S. de Carvalho, Vinícius E. Pimentel, Malena M. Pérez, Camilla N. S. Oliveira, Lilian C. Rodrigues, Victor A. F. Bastos, Sandra O. C. Tella, Ronaldo B. Martins, Augusto M. Degiovani, Fátima M. Ostini, Marley R. Feitosa, Rogerio S. Parra, Fernando C. Vilar, Gilberto G. Gaspar, José J. R. da Rocha, Omar Feres, Eurico Arruda, Sandra R. Maruyama, Elisa M. S. Russo, Angelina L. Viana, Isabel K. F. M. Santos, Vânia L. D. Bonato, Cristina R. B. Cardoso, Jose E. Tanus-Santos, Eduardo A. Donadi, Lucia H. Faccioli, Marcelo Dias-Baruffi, Ana P. M. Fernandes, Raquel F. Gerlach, Carlos A. Sorgi

**Affiliations:** 1Departamento de Análises Clínicas, Toxicológicas e Bromatológicas, Faculdade de Ciências Farmacêuticas de Ribeirão Preto-FCFRP, Universidade de São Paulo-USP, Ribeirão Preto 14040-903, Brazil; pedrovieira_sn11@hotmail.com (P.V.d.S.-N.); cafuzo@usp.br (C.A.F.); dianamota.t@gmail.com (D.M.T.); jonatancarvalho@usp.br (J.C.S.d.C.); viniciuspimentel@usp.br (V.E.P.); malenac@usp.br (M.M.P.); camilla.narjara@outlook.com (C.N.S.O.); liktaldi@fcfrp.usp.br (L.C.R.); victor_ldn@hotmail.com (V.A.F.B.); elisa@fcfrp.usp.br (E.M.S.R.); cristina@fcfrp.usp.br (C.R.B.C.); faccioli@fcfrp.usp.br (L.H.F.); mdbaruff@fcfrp.usp.br (M.D.-B.); 2Programa de Pós-Graduação em Imunologia Básica e Aplicada-PPGIBA, Instituto de Ciências Biológicas, Universidade Federal do Amazonas-UFAM, Manaus 69080-900, Brazil; 3Departamento de Biologia Básica e Oral, Faculdade de Odontologia de Ribeirão Preto, Universidade de São Paulo-USP, Ribeirão Preto 14040-904, Brazil; valeriab.valle@usp.br (V.B.d.V.); pabasile@usp.br (P.A.B.); 4Departamento de Enfermagem Geral e Especializada, Escola de Enfermagem de Ribeirão Preto-EERP, Universidade de São Paulo-USP, Ribeirão Preto 14040-902, Brazil; talitafernandes@usp.br (T.M.F.); angelina.lettiere@usp.br (A.L.V.); anapaula@eerp.usp.br (A.P.M.F.); 5Departamento de Bioquímica e Imunologia, Faculdade de Medicina de Ribeirão Preto-FMRP, Universidade de São Paulo-USP, Ribeirão Preto 14040-900, Brazil; thaisfragasilva@usp.br (T.F.C.F.-S.); imsantos@fmrp.usp.br (I.K.F.M.S.); vlbonato@fmrp.usp.br (V.L.D.B.); 6Departamento de Química, Faculdade de Filosofia, Ciências e Letras de Ribeirão Preto-FFCLRP, Universidade de São Paulo-USP, Ribeirão Preto 14040-901, Brazil; 7Departamento de Farmacologia, Faculdade de Medicina de Ribeirão Preto-FMRP, Universidade de São Paulo-USP, Ribeirão Preto 14040-900, Brazil; conde@fmrp.usp.br (S.O.C.T.); tanus@fmrp.usp.br (J.E.T.-S.); 8Departamento de Biologia Celular e Molecular e Bioagentes Patogênicos, Faculdade de Medicina de Ribeirão Preto-FMRP, Universidade de São Paulo-USP, Ribeirão Preto 14040-900, Brazil; ronaldobmjunior@gmail.com (R.B.M.); eaneto@fmrp.usp.br (E.A.); 9Hospital Santa Casa de Misericórdia de Ribeirão Preto, Ribeirão Preto 14085-000, Brazil; augustomd@msn.com (A.M.D.); tata_ostini@hotmail.com (F.M.O.); 10Departamento de Cirurgia e Anatomia, Faculdade de Medicina de Ribeirão Preto-FMRP, Universidade de São Paulo-USP, Ribeirão Preto 14040-900, Brazil; mrfeitosa@hcrp.usp.br (M.R.F.); rsparra@hcrp.usp.br (R.S.P.); jjrocha1@bol.com.br (J.J.R.d.R.); omar.feres2021@gmail.com (O.F.); 11Departamento de Clínica Médica, Faculdade de Medicina de Ribeirão Preto-FMRP, Universidade de São Paulo-USP, Ribeirão Preto 14040-900, Brazil; fcvilar@gmail.com (F.C.V.); ggaspar@hcrp.usp.br (G.G.G.); eadonadi@fmrp.usp.br (E.A.D.); 12Departamento de Genética e Evolução, Centro de Ciências Biológicas e da Saúde, Universidade Federal de São Carlos-UFSCar, São Carlos 13565-905, Brazil; sandrarcm@ufscar.br

**Keywords:** metalloproteinases, sHLA-G, sTREM-1, lipid peroxidation, COVID-19

## Abstract

Patients with COVID-19 predominantly have a respiratory tract infection and acute lung failure is the most severe complication. While the molecular basis of SARS-CoV-2 immunopathology is still unknown, it is well established that lung infection is associated with hyper-inflammation and tissue damage. Matrix metalloproteinases (MMPs) contribute to tissue destruction in many pathological situations, and the activity of MMPs in the lung leads to the release of bioactive mediators with inflammatory properties. We sought to characterize a scenario in which MMPs could influence the lung pathogenesis of COVID-19. Although we observed high diversity of MMPs in lung tissue from COVID-19 patients by proteomics, we specified the expression and enzyme activity of MMP-2 in tracheal-aspirate fluid (TAF) samples from intubated COVID-19 and non-COVID-19 patients. Moreover, the expression of MMP-8 was positively correlated with MMP-2 levels and possible shedding of the immunosuppression mediator sHLA-G and sTREM-1. Together, overexpression of the MMP-2/MMP-8 axis, in addition to neutrophil infiltration and products, such as reactive oxygen species (ROS), increased lipid peroxidation that could promote intensive destruction of lung tissue in severe COVID-19. Thus, the inhibition of MMPs can be a novel target and promising treatment strategy in severe COVID-19.

## 1. Introduction

Severe acute respiratory syndrome coronavirus 2 (SARS-CoV-2) is the infectious agent that causes COVID-19. So far, a massive number of people worldwide have died from severe forms of COVID-19, and vaccination was associated with a significant reduction in COVID-19 infection as well as a reduction in mortality [1]. The severe forms of COVID-19 screening a systemic inflammatory response, thromboembolic complications, and multi-organ failure [2]. Often, the lung becomes non-functional, with potentially fatal consequences [3]. This disease, still today, is a challenge for intensive care physicians, which treat the severe forms of this disease with supportive treatment only [4].

Overall, the pathological mechanisms that lead to death in severe forms of COVID-19 remain unclear. Nonetheless, an important point of this analysis is the fact that the severity of COVID-19 is exacerbated by pre-existing comorbidities, such as hypertension, heart disease, obesity, diabetes, cancer, and a compromised immune system [5,6]. Among comorbidities, all of them, except the last one, are known to show increased levels of gelatinases in plasma [7]. Therefore, it is tempting to hypothesize that the pre-infection level of plasma matrix metalloproteinases (MMPs) or the potential of the host cells to secrete these proteases (depending on age and genetic polymorphisms) and the activation of those in the host would result in a worsening of the COVID-19 disease course. In many scenarios, excessive proteolysis also involves the activation of several other proteinases, such as coagulation or complement protease cascades [8].

MMPs are believed to be contributing factors to injurious processes in lung pathologies [9], and recent clinical data suggest an increase in plasma MMPs levels in patients with COVID-19 [10]. MMPs represent a family of proteolytic enzymes that contain a zinc ion at the active site of catalysis [11,12] and can cleave a wide variety of substrates, including extracellular matrix (ECM) components (collagens, fibronectin, and elastin), secreted and ECM-anchored growth factors, chemokines, and cytokines [13]. In this regard, some evidence denotes that the soluble form of the human leukocyte antigen-G (HLA-G), a key membrane molecule in pregnancy immune regulation and inflammation control [14], and the Triggering receptor expressed on myeloid cells 1 (TREM-1), a member of the immunoglobulin superfamily expressed on myeloid and epithelial cells [15] is generated through the shedding by MMPs pathways [15,16,17,18].

The hyper-inflammatory responses to COVID-19 are characterized by cytokine release syndrome (also called “cytokine storm”), leading to acute respiratory distress syndrome (ARDS), increased pulmonary edema, fibrosis, and hypoxia [19,20]. In addition, excessive neutrophil recruitment into the alveolar space can cause lung injury, as neutrophils release large amounts of proteases, reactive oxygen species (ROS), and extracellular neutrophil extracellular traps (NETs), which are detrimental to host tissues [21]. Autopsy tissue examination of critically ill patients with COVID-19 shows a high degree of diffuse alveolar damage, perivascular inflammatory cell infiltration [22], extensive damage to the vascular lining [23,24], and enhanced remodeling of the fibrosis and extracellular matrix (ECM) in the lung [25].

Inflammatory and parenchymal cells perform some of their functions by releasing MMPs into the lung [26]. The “imbalanced” or excessive proteolytic activity characteristic of the time when active MMPs appear in tissues or plasma is generally found in exacerbated inflammatory responses and results in excessive destruction of structure [27]. Therefore, MMPs are crucial components of the processes leading to the state of COVID-19 pneumonia. To better define the role of MMPs in severe COVID-19 lungs, we investigate the expression of MMPs in the lung parenchyma and the association between the bronchoalveolar-tracheal levels and activity of MMP-2 and MMP-8 to the immune response associated with sHLA-G and sTREM-1 release, in addition to tissue damage by oxidative stress in COVID-19 outcomes.

## 2. Materials and Methods

### 2.1. Ethical Approval

All participants or legal tutors gave their written consent through the informed consent form, in accordance with the regulations and human ethics guidelines of the National Council on Human Research (CONEP) and the Research Ethics Committee from Faculdade de Ciências Farmacêuticas de Ribeirão Preto da Universidade de São Paulo (CEP-FCFRP-USP). The research protocol was approved and received the Certificate of Ethics Presentation and Appreciation (CAAE: 30525920.7.0000.5403). The sample size was determined by the convenience of collection, availability in partner hospitals, participation agreement, and pandemic conditions within the local community.

### 2.2. Study Design and Participants

This observational, analytical, and prospective study was conducted at the Hospital Santa Casa de Misericordia of Ribeirão Preto and Hospital São Paulo of Ribeirão Preto-Brazil from June 2020 to January 2021, using stringent and reasonable inclusion and exclusion criteria: adults who tested positive for COVID-19 and controls who tested negative for COVID-19; exclusion for children under 18 years of age and pregnant or lactating women. In total, non-COVID-19 subjects (*n* = 13), who were hospitalized and intubated due to different clinical primary conditions (Appendix A) and negative for SARS-CoV-2 infection, along with patients in severe/critical illness (*n* = 39), intubated and hospitalized in intensive care unit (ICU) who tested positive for SARS-CoV-2 infection, as determined by analyzing nasopharyngeal swabs using a genomic RNA assay with RT-PCR (Biomol OneStep Kit/COVID-19-Instituto de Molecular Biology of Paraná-IBMP Curitiba/PR-Brazil). Peripheral blood samples were collected by venipuncture on first admission and/or during hospitalization for clinical analysis.

### 2.3. Data Collection

The data were collected from the electronic medical record systems of each patient and carefully revised. We included socio-demographic information, comorbidities, medical history, clinical symptoms, routine laboratory tests, clinical interventions, and outcomes. Data collection from laboratory results was defined by considering the first examination at admission (within 24 h of admission) as the primary endpoint. Blood exams of hospitalized patients were performed at clinical analysis laboratories in their respective hospitals.

### 2.4. Tracheal Aspirate Fluid (TAF) Collection and Processing

TAF samples were collected from hospitalized patients with severe COVID-19 and non-COVID-19 (control), as previously described [28], using a catheter (Mark Med, Porto Alegre, Brazil) with a Trach Care closed endotracheal suction system (Bioteque Corporation, Chirurgic Fernandes Ltd., Santana Parnaíba, Brazil), and sterile polypropylene vials (Biomeg-Biotec Hospital Products Ltd., Mairiporã, Brazil), under aseptic conditions. Approximately 5–10 mL of sterile isotonic saline was instilled into the endotracheal tube; the individual was manually ventilated for 3 breaths, and the trachea was suctioned twice, each time for 5 s; and the TAF samples were collected, placed on ice, and processed within 4 h. In a Biosafety Level 3 facility at Departamento de Bioquímica e Imunologia, Faculdade de Medicina de Ribeirão Preto, Universidade de São Paulo, the TAF samples were placed in 15 mL tubes and diluted with 0.1 M phosphate buffered saline (PBS) (2:1, *v*/*v*). After centrifugation (700× *g*/10 min), the sample supernatants were recovered and stored at −80 °C and further used for inflammatory mediators and MMPs analysis. Red blood cells in the centrifuged pellet were lysed with 1 mL of NH_4_Cl buffer (0.16 M) for 5 min. The remaining cells were washed with 10 mL of PBS and suspended in 2% heat-inactivated fetal bovine serum (FBS) in PBS. Cell suspension aliquots were diluted with Trypan blue and counted in an automated cell counter (Countess, Thermo Fisher Scientific, Waltham, MA, USA). The leukocyte numbers were adjusted to 1 × 10^9^ cells/L for differential counts.

Subsequently, differential leukocyte counts (mononuclear cells, neutrophils, and eosinophils) were performed by adding 100 µL of the TAF cell sample to cytospin and stained with Fast Panoptic dye (Laborclin-Laboratory Products Ltd., Pinhais, Brazil). An average of 200 cells were examined and morphologically characterized under an optical microscope (Zeiss EM109; Carl Zeiss AG, Oberkochen, Germany) equipped with a 100× objective (immersion oil) attached to a digital camera (Olympus Soft Imaging Solutions Gmbh, Germany) and analyzed with ImageJ software (1.45 s) (National Institutes of Health, Rockville, MD, USA).

### 2.5. Soluble TREM-1 and MMPs Quantification

sTREM-1, MMP-2, and MMP-8 levels were measured in TAF samples using an ELISA kit (DuoSet-Human TREM-1, DuoSet-Human Total MMP-2, and DuoSet-Human Total MMP-8, R&D System, Minneapolis, MN, USA) according to the manufacturer’s specifications.

### 2.6. Quantification of Active MMPs by Zymography

The active MMP-2 and MMP-9 forms were measured in TAF samples by gelatin zymography as previously described [29]. Briefly, the samples were diluted (1:5 *v*/*v*) with extraction buffer containing 10 mM CaCl_2_, 50 mM Tris-HCl pH 7.4 The protein content was measured using the Bradford method [30]. Protein concentration varied from 3 to 40 μg/μL. Other aliquots with the same protein concentration were then used for total protein analysis in conventional SDS-PAGE and zymography gels. 10 μg of protein from each sample was mixed with non-reducing sample buffer (see below) and used for conventional gel electrophoresis (the gel did not contain gelatin), followed by silver staining. The samples were then further diluted to run 1 μg of protein/lane for the zymograms. Samples were always kept on ice and gels were run on ice. Prior to running, samples in non-reducing sample buffer (2% SDS, 125 mM Tris-HCl, pH 6.8, 10% glycerol, and 0.001% bromophenol blue) were kept 5 min. at 60 °C in a bath to reduce dymers. SDS-PAGE gels (12%) co-polymerized or not with 1% gelatin and casein were used for this study. Casein and gelatin gels were used in the initial tests, and the results are shown in the Appendix A. After the electrophoresis was complete, the gels were incubated twice for 30 min at room temperature in 2% Triton X-100 solution, followed by incubation at 37 °C for 18 h in Tris-HCl buffer, pH 7.4, containing 10 mM CaCl_2_. The gels were stained with 0.05% Coomassie brilliant blue G-250 for 18 h and unstained with 30% methanol and 10% acetic acid. Gelatinolytic activities were detected as unstained bands against the background of Coomassie blue-stained gelatin. Enzyme activity was assayed by densitometry using an electrophoresis documentation system (ChemiDoc MP Imaging System, BioRad, Hercules, CA, USA). Enzyme activities were normalized against an internal standard (fetal bovine serum, FBS) run in all gels, to allow inter-gel analyses and comparisons. The molecular weight for pro-MMP-2/MMP-2 (72–64 kDa) and pro-MMP-9/MMP-9 (92–86 kDa), respectively, by the relation of logMr to the relative mobility of the SDS-PAGE protein ladder (BlUeye Prestained Protein Ladder-Sigma, Saint Louis, MO, USA). The results were then scanned at 400 dpi and analyzed with ImageJ software (1.45 s) (National Institutes of Health, Rockville, MD, USA).

### 2.7. Soluble HLA-G Quantification

sHLA-G levels in TAF samples were measured using a sandwich ELISA with mAb anti-HLA-G (MEM-G/9-Exbio, Czech Republic) and anti-β2-microglobulin (Dako, Brooklyn, NY, USA) as capture and detection antibodies, respectively [31]. Briefly, microtitration plates were coated with MEM-G/9 (10 μg/mL) at 4 °C for 18 h. After blocking unspecific ligands in the wells with 300 µL of diluent (Dako, Brooklyn, NY, USA) for 2 h, two-fold diluted TAF samples (50 µL) were added and incubated for 2 h. The wells were then incubated with rabbit-anti-human β2-microglobulin detection antibody (Dako, Brooklyn, NY, USA) for an additional hour. 100 μL of horseradish peroxidase enhancer (Dako, Brooklyn, NY, USA) was then added and incubated for 1 h. All incubation steps were performed at room temperature. Each step was followed by 4 washes using a specific washing buffer containing PBS and 0.1% Tween (Sigma, Saint Louis, MO, USA). Finally, the wells were incubated with substrate (tetramethylbenzidine-TMB) in the dark for 30 min. After the addition of 1 N HCl, optical densities were measured using microplate reader (SpectraMax Plus 384, Molecular Devices, San Jose, CA, USA), applying an absorbance filter of 450 nm. All samples were assayed in duplicate, and the total levels of sHLA-G were determined from a standard curve of five points, using calibrated HLA-G5 dilutions as a standard. Results were expressed as ng/mL.

### 2.8. Assessment of Lipid Peroxide Levels (MDA)

Lipid peroxide levels in TAF samples were determined by measuring thiobarbituric acid-reactive substances (TBARS) using a fluorimetric method described previously [32,33]. In this method, malondialdehyde (MDA) reacts with thiobarbituric acid (TBA) under high temperatures (90–100 °C) and acidic conditions, generating the MDA-TBA adduct. The MDA-TBA adduct was determined fluorometrically at an excitation wavelength at 515 nm and emission at 553 nm and uses 1,1,3,3-tetramethoxypropane as standard for curve. All the measurements were performed using the Synergy 2 multi-mode microplate reader and Gen5 Software (BioTek, Winooski, VT, USA). Briefly, TAF samples were transferred to an equal volume of 20% (*v*/*v*) cold trichloroacetic acid in 0.6 M HCl, mixed and centrifuged at 1200× *g* for 15 min. In a volume of clear supernatant, a 0.2 volume of 0.12 M thiobarbituric acid/0.26 M Tris, pH 7.0 was added, and immersed in a boiling water bath for 1 h. The lipid peroxide levels were expressed in terms of MDA (nmol/mL).

### 2.9. Detection of SARS-CoV-2 RNA

Total RNA was extracted from 20 uL of TAF samples with TRIzol reagent according to the manufacturer’s instructions. The qPCR assay was performed for the E gene region and N2 region in the N gene [34,35]. Real-time RT-PCR was performed using TaqPath 1-Step qRT-PCR Master Mix (Applied Biosystems, Foster City, CA, USA) on a StepOne Plus real-time PCR system (Applied Biosytems, Foster City, CA, USA) with the following parameters: 25 °C for 2 min, 50 °C for 15 min, 95 °C for 2 min, followed by 45 cycles of 94 °C for 5 s and 60 °C for 30 s. SARS-CoV-2 viral loads were determined using a standard curve prepared with a plasmid containing all three targets for the sets of primers/probes designed by the CDC protocol (N1, N2 and N3) [36].

### 2.10. Statistical Analysis

Data are presented in tables and graphs, using GraphPad Prism^TM^ software, version 9 (San Diego, CA, USA). Taking into account the nonparametric distribution of the data, comparative analyzes between groups were performed using the Mann-Whitney or Kruskal-Wallis tests, with *p* < 0.05 significant. The dependence on multiple variables was calculated using Spearman’s correlation test, and the differences were considered statistically significant with *p* < 0.05.

### 2.11. Re-Analysis of Proteomics Data

Protein expression data for lung biopsies from COVID-19 and non-COVID-19 samples were obtained from the Appendix A of an original proteomic study [37]. Only proteins with expression values in at least three samples for each group were used in the statistical analysis. Student’s test between the COVID-19 and non-COVID-19 groups was carried out for all selected proteins after transforming expression values using the Log2 scale, and the false discovery rate was controlled using the Benjamini and Hockberg correction [38]. The results of MMPs were extracted from the whole statistics using both nominal and adjusted *p*-values with significant results (*p* < 0.05) (Appendix A). Statistical analysis was performed in R [39], and figures were produced using the package ggplot2 [40].

## 3. Results

### 3.1. Participants Demographic Data and Clinical Characteristics

In total, 39 patients with laboratory-confirmed COVID-19 were enrolled in this study, classified as severe/critical ill with oxygen support, intubated in ICU hospital care, in addition to 13 critical volunteers, negative for COVID-19 (non-COVID-19) but hospitalized and intubated due to different primary clinical conditions (Appendix A). The median age and BMI were not significantly different between non-COVID-19 volunteers and patients with COVID-19. The percentage of men was higher among patients with COVID-19 than in non-COVID-19, as well as the percentage of comorbidities—especially hypertension—with significant consideration (Table 1). The most common symptoms in the patients were dyspnea, cough, fever, and myalgia followed by diarrhea, anosmia, and dysgeusia; in non-COVID-19, only dyspnea was observed (Table 1). Moreover, the absolute blood counts of erythrocytes and hemoglobin concentration in the COVID-19 patient group were different from those in the non-COVID-19 group, while the counts of leukocytes, neutrophils, lymphocytes, neutrophil/lymphocytes ratio (NLR), monocytes and platelets were not significantly different between those groups (Table 1).

The median time of hospitalization for COVID-19 patients was 6.5 days and 23 days for non-COVID-19. Additionally, all those hospitalized patients received respiratory support by invasive mechanical ventilation, but the oxygen saturation was significantly lower among COVID-19 patients (91%) as compared to the non-COVID-19 group (94%) (Table 1). The percentage of death outcomes was 84.6% for COVID-19 patients and 61.5% for non-COVID-19 individuals (Table 1).

### 3.2. Elevated Expression of MMPs on Lung-Tissue and MMP-2 Active form on TAF Samples from COVID-19 Patients

Initially, we reanalyzed proteomics data from lung biopsy samples of COVID-19 patients (*n* = 30) and non-COVID-19 individuals (*n* = 6) [37]. Comparative analysis was performed using the log2 target of MMP protein expression: MMP-1, MMP-2, MMP-7, MMP-8, MMP-9, MMP-10, MMP-12, MMP-14, MMP-15, MMP-19, MMP-28, MMP-23B, and MMP-24OS (Figure 1A). This target proteomics approach indicated that MMP-2, MMP-7, MMP-8, and MMP-14 figured out the COVID-19 perturbation with a significant increase compared to non-COVID-19 individuals. However, the expression of the MMP-15 protein in COVID-19 was lower than in non-COVID-19 lung tissue. Therefore, we hypothesize that MMPs were strongly associated with lung COVID-19 severity.

To verify the importance of MMPs in COVID-19 lung disease pathogenesis, we performed a protease screening on TAF samples from our cohort. For this, we focused on the classical protease cleavage of substrates in gels (zymograms), including gelatin and casein as substrates co-polymerized in SDS-PAGE gels. Results indicated a strong gelatinolytic activity and a much lower caseinolytic activity when the same amount of TAF–COVID-19 protein was used in the gels (Appendix A).

Next, the gelatinolytic activity in zymograms was observed after the gels had been incubated overnight in the presence of protease inhibitors. The control gel was not incubated with proteinase inhibitors (Appendix A), while other gels were incubated with the inhibitors of the most common classes of protease found in eucariotic cells, namely: serine-, metallo-, and cysteine-proteinases. The gelatin-containing zymograms had been prepared and run in the same way and contained the same samples as the gel presented in Appendix A. Appendix A shows the gel incubated with 1 mM 1-10-Phenanthroline, a preferential zinc-chelator, and a potent metalloproteinase inhibitor. Appendix A shows the gel incubated with 1 mM Phenylmethylsulphonylfluoride (PMSF), a serine-proteinase inhibitor (PMSF-gel). The gel incubated with N-ethyl-maleimide, a cysteine-protease inhibitor, is not shown. No significant activity was observed. When considering that 1 μg of total TAF protein was loaded onto the lanes containing the higher masses, it can be easily realized that a huge concentration of gelatinases is present in the TAF samples (at this point, the samples were selected at random, since these experiments were part of the initial characterization steps). Regarding the use of inhibitors, strong gelatinolytic activity was observed in the control gel, with a similar result for gels incubated with PMSF and NEM, but without detected gelatinolytic activity in gel incubated with 1-10-Phenanthroline. Interestingly, neither serine nor cysteine proteinases were detected in our TAF samples in significant amounts, even using very high amounts of protein (considering that zymograms regularly detect nanograms of gelatinases and therefore the usual amounts of load used for Western Blots (usually 30 μg), for example, cannot be applied to zymograms. Moreover, it became clear that metalloproteinases were the most active proteases in our TAF samples, and second protein quantification was performed, to ensure that no problems in comparison of samples would come from the serial dilution of samples, that was necessary to apply 0.1 μg of total protein per sample/lane. This was necessary since quantification of gelatinolytic bands should only be done when the gelatinotic bands are found in a range where the total activity was still not reached (for plasma, we used 0.2 to 1 μL to obtain gelatinolytic bands that were adequate for quantification and comparison between patients).

As demonstrated in Figure 1B-panel and Appendix A, the zymogram showed the presence of complex, pro- and active forms of MMPs. Indeed, pro-MMP-2 and pro-MMP-9 were present in all TAF samples from non-COVID-19 and COVID-19 patients. However, activated MMP-2 was prominent on COVID-19 samples only, while activated MMP-9 was not found, neither in COVID-19 or non-COVID-19 samples. Regarding stratifying the COVID-19 patients based on the outcome (survival and non-survival), we observed that the ratio of active-/pro-MMP-2 levels was significantly increased in the group of non-survival patients (Figure 1B-graphic). MMP-9 levels were not associated with COVID-19 lung pathology and did not show a correlation with mechanical ventilation in non-COVID-19 subjects.

### 3.3. MMP-2 and MMP-8 Expression Increased in Lung from Patients with Non-Survival COVID-19 and Was Correlated with the Release of sTREM-1 and sHLA-G

Considering the total protein form by ELISA, MMP-2 and MMP-8 were significantly higher in TAF from patients with COVID-19 compared to non-COVID-19 (Figure 2A). Furthermore, comparing non-survival with survival COVID-19 patients, we observed a significant increase in MMP-2 and MMP-8 levels in the non-survival group, compared to the survival group and the non-COVID-19 group. There was no statistical difference between the COVID-19 survival patients to non-COVID-19 (Figure 2A).

MMP-2 and MMP-8 were unambiguously elevated proteinases in the lung of patients with COVID-19. To date, those MMPs have been only analyzed in systemic COVID-19 studies [10,41]. On the other hand, there are some key proteins for the immune response that can be generated through proteolytic cleavage, which is known to be mediated by MMPs, such as sHLA-G [18] and sTREM-1 [15]. Taken together, our data demonstrated that both sHLA-G and sTREM-1 levels on TAF samples were elevated in COVID-19 patients and sTREM-1 positively correlated with MMP-8, while sHLA-G levels positively correlated with both MMP-2 and MMP-8 expression (Figure 2B). However, some evidence suggested that TREM-1 site cleavage to release sTREM-1 was specific for MMP-8 activity [15] and HLA-G site cleavage to release sHLA-G was specific for MMP-2 activity [42]. In fact, we confirmed the MMP-8 specific axis to sTREM-1, but there was an ambiguity about MMPs and sHLA-G release in our data. For this, we analyzed a Spearman test correlation between MMP-2 and MMP-8 levels and showed a positive and significant result on TAF samples (Figure 2C). These data suggested that MMP-8 could perform to generate active-MMP-2, and MMP-2 was involved in sHLA-G release (Figure 2D).

### 3.4. Relationship of MMP-2 Levels and Oxidative Stress in the Lung of Non-Survival COVID-19 Patients

Reactive oxygen species (ROS) disrupt lipids, proteins, and DNA, potentially resulting in tissue damage and cell death [43]. The interaction of ROS with cell membranes leads to the generation of lipid peroxides, which can be quantified using thiobarbituric reactive substances (TBARS) and could represent oxidative stress [32]. Although nonspecific, TBARS are considered a stable marker of free radical damage and oxidative stress due to their rapid generation and excretion. We showed in Figure 3A that the level of lipid peroxidation (MDA) was significantly increased in TAF samples of COVID-19 compared to non-COVID-19. Additionally, when we stratified the COVID-19 patients in the survival and non-survival group, we observed that this high level of lipid peroxidation was significantly related to the non-survival patients, compared to survival patients and non-COVID-19 (Figure 3A).

As previously demonstrated, neutrophils were the main cells whose increases in the blood of count patients with severe COVID-19 [44], and this phenomenon can be associated with hyper-inflammatory response and “cytokine storm” [45]. In fact, neutrophilia is an indicator of severe respiratory symptoms and an unfavorable outcome in COVID-19 [46]. We next determined the cell infiltration in TAF samples (Appendix A), and we detected a significant increase only in neutrophil counts from patients with COVID-19 compared with non-COVID-19. In this context, in Figure 3B we demonstrated that the neutrophils count in non-survival COVID-19 patients was significantly higher compared to non-COVID-19. The regression approach revealed a positive association of lipid peroxidation levels (MDA) and neutrophil counts in TAF samples, suggesting that ROS species production and consequently oxidative stress could be related to these immune cells in the lung microenvironment of SARS-CoV-2 infection (Figure 3B).

Interestingly, we showed a positive and significant correlation between lipid peroxidation (MDA) levels and MMP-2 expression on TAF samples (Figure 3C). We emphasized a relative looping in our data when ROS formation by oxidative stress, probably due to neutrophils activity, could trigger MMP-2 activation and MMP-2 function could increase ROS formation in the lung of COVID-19 patients (Figure 3D).

## 4. Discussion

Survival of COVID-19 patients with severe symptoms depends on the extension of lung injury, damage to other organs, comorbidities of the infected patient, and appropriate viral immune response [47]. Even though MMPs play a key role in lung immunity by facilitating the influx of inflammatory cells and modulating the activities of inflammatory mediators and defensins [9], the unbalanced levels of a variety of MMPs have been predictable in many lung disorders [9]. In this context, cytokines, inflammatory mediators and MMPs levels could define an immune-based biomarker system of COVID-19. However, no enzyme activity of MMPs has been studied in the lungs of patients with COVID-19 and correlation to tissue pathology. We demonstrated, for the first time, that the enzymatic activity and levels of MMP-2 and MMP-8 increased significantly in the lung microenvironment of intubated patients with COVID-19, and this MMP-axis was associated with infiltration of lung neutrophils, oxidative stress and release of sTREM-1 and sHLA-G, important mediators for the regulation of immune response.

As in all tissues, the expression of MMPs in the lung is a highly regulated process [48,49]. Moreover, MMPs degrade the ECM of the interstitium leading to an increase in alveolar permeability that is observed in destructive lung diseases, including ARDS, COPD, tuberculosis, sarcoidosis, and idiopathic pulmonary fibrosis (IPF) [50,51]. Although small amounts of MMP-2 and MMP-14 are present in the lung lining fluid under normal conditions, other MMPs, such as MMP-7, MMP-8, MMP-9, and MMP-12, are up-regulated under many pathological conditions [48,52,53]. Indeed, epithelial cells from bronchoalveolar lavage fluid (BALF) on severe COVID-19 showed elevated frequencies of MMP-7^+^ and MMP-9^+^ and a tendency to increase portions of MMP-2^+^ and MMP-13^+^ compared to mild cases [54]. It appears partly reasonable with our results from proteomics reanalysis, which showed a significant increase in detection of MMP-2, MMP-7, MMP-8, and MMP-14, but not the expression of the MMP-9 protein, in lung tissue from COVID-19 compared to non-COVID-19 subjects.

MMPs are initially synthesized in a latent pro-form as zymogens. The basic structure of the catalytic portion of proteases consists of a catalytic domain with three histidine residues related to a zinc atom (Zn^2+^) and a pro-domain containing a cysteine [55]. For enzymatic activation, proteolytic cleavage of the pro-domain and exposition of the catalytic site are required [56]. MMP-2 was synthesized by a wide variety of cells, including fibroblasts, endothelial cells, and alveolar epithelial cells and plays role in lung inflammatory diseases [57,58]. In fact, the absence of MMP-2 was protective in allotransplant models reducing cellular infiltration and fibrosis; in contrast, deficiency in MMP-2 increased the susceptibility of mice to lethal asphyxiation in an asthma model [59,60]. We observed that TAF samples from COVID-19 and non-COVID-19 individuals expressed high levels of pro-MMP-9 and pro-MMP-2 by zymogram. However, only the active form of MMP-2 was extremely associated with non-survival COVID-19 patients. Post-translational modification of MMPs in the lung could be a crucial step in regulating the action of MMPs in vivo. Several MMPs have soluble and cell surface forms, providing another level of regulation through compartmentalization/localization [61]. In addition, endogenous families of inhibitory proteins, TIMP1-4 and α2-macroglobulin, are known to regulate MMPs at the post-translational level [62]. Uncontrolled MMP-2-activity can be highly pro-inflammatory and affect lung physiology with severe COVID-19.

Apparently, no single MMP is a fundamental mediator of any specific pulmonary pathology, as each MMP plays an individual role in specific periods and potential functional redundancy, based on the numerous overlapping substrate molecules that exist at the site of activity of MMPs [9]. Indeed, the TAF samples of COVID-19 patients presented a high amount of MMP-8, such as high levels of MMP-2. The expression of both MMP was a significant contributor to non-survival COVID-19 patients. Compartmentalization of MMPs in inflammatory cells is another mechanism of regulation; for instance, pro-MMP-8, pro-MMP-9, and pro-MMP-25 are packaged into peroxidase-negative granules within neutrophils to be released upon leukocyte activation [63]. Likewise, neutrophils exposed to pro-inflammatory cytokines increase the expression of active MMP-8 on the cell surface, and its localization on the cell surface confers resistance to inhibition with TIMP [61]. In accordance with our work, it was characterized by immunoblotting MMP-2, -8, and -9 and TIMP-2 in TAF samples from preterm infants with respiratory distress during the first postnatal days, suggesting that in preterm infants, increased pulmonary MMP-8 levels participate in the acute inflammatory injury [64].

In response to SARS-CoV-2 infection, the host’s immune system and target cells are likely to release MMPs [10,65,66]. MMP activity normally governs the release of substrates that are anchored either at the extracellular matrix or cell membrane, such as growth factors and cell membrane receptors [67,68]. HLA-G is a non-classical HLA class I antigen, which is pondered as an immune inhibitory mediator and can be up-regulated by several viral infections, including SARS-CoV-2, which can render comprehensive immunosuppressive roles in favoring virus immune evasion and subsequent disease progression [69,70]. Soluble HLA-G proteins can be generated through proteolytic release; for example, there is an effective link between MMP-2 and the shedding of HLA-G, but not for MMP-9 in this process [42]. TREM-1, another membrane receptor, amplifies the pro-inflammatory response in synergism to Toll-like receptors, which recognize a wide range of bacterial, fungal, and viral components [71]. Another study reported an MMPs cleavage site within the TREM-1 sequence and demonstrate a correlation to MMP-9 activity [72]. However, our group demonstrated that plasma expression of MMP-8 was positively correlated with sTREM-1 levels, specifically in the group of patients with severe COVID-19 [41]. As expected, in our TAF samples, the MMP-8 expression correlated positively with sTREM-1 production. However, the release of sHLA-G was positively correlated with both MMP-2 and MMP-8 levels.

Classically, MMP-2 activation on the cell surface is critically dependent on the binding of pro-MMP-2 to MMP-14 and TIMP-2. This complex allows a second active MMP-14 to cleave the pro-domain and release the active-MMP-2 [73]. On the other hand, MMP-7 was able to activate pro-MMP-8, and the accumulation of pro-MMP-8 in the absence of MMP-7 was accompanied by a decrease in pro-MMP-13 levels, suggesting the interaction between MMP-7, MMP-8, and MMP-13 to regulate collagen turnover [74]. However, in TAF samples, we observed a positive and significant correlation between MMP-2 and MMP-8 levels. Since we had evidence of increased expression of MMP-14 and MMP-7 in COVID-19 lung tissue by proteomics reanalysis, both described activation pathways that could be effective in TAF samples to produce active-MMP-2 and active-MMP-8. On the other hand, we could have a new activation positive looping intricate MMP-8 and MMP-2, and this phenomenon could justify the effective action of MMP-2 on HLA-G shedding and indicated that the positive correlation of MMP-8 expression to sHLA-G levels was an indirect axis linked to MMP-2 activation.

Increased inflammatory mediators activate neutrophils and alveolar macrophages, which liberate MMPs and oxygen radicals, thus producing more lung damage, increasing vascular leakage and cell apoptosis [75]. Oxidative stress results from the overproduction or inhibited inactivation of reactive oxygen species (ROS), causing alterations in the redox state of proteins and lipids [76]. Increased ROS formation triggers protein oxidation and activation of a cascade of cell signaling events, resulting in endothelial dysfunction and MMPs activation [77,78]. It has been shown that changes in the tissue concentrations of O^2−^, H_2_O_2_, and ONOO^−^ affect MMP-2 activity [77,79]. However, the increased MMP-2 activity in the vascular system could directly activate pro-oxidant pathways, for example, MMP-2 cleaved pro-heparin binding epidermal growth factor (HB-EGF) and the soluble HB-EGF bind the EGF receptor (EGFR), downstream NADPH oxidase, which increased ROS formation [80]. Regarding this, the activation of MMP-2 and MMP-9 was directly involved in the vascular remodeling observed in hypertension [81]. We observed an increased lipid peroxidation levels (MDA quantification) on TAF samples from COVID-19 patients compared to non-COVID-19, and this oxidative stress was significantly more prominent in no-survival than survival patients. In this context, neutrophil counts were higher in samples from COVID-19 patients, indicating a positive correlation between neutrophil infiltration and lipid peroxidation levels (also, ROS production). Moreover, MMP-2 levels were positive and significantly correlated with lipid peroxidation concentration, suggesting another positive looping of neutrophils producing ROS species; ROS triggers MMP-2 activation and MMP-2 enhanced oxidative stress in the COVID-19 lung.

Thus, despite the fact that great efforts have been made toward the development of inhibitors of MMPs, it is not clear whether MMP inhibition is beneficial or harmful in diseases. COVID-19 patients with severe symptoms exhibit impaired endothelial and microcirculatory functions, neutrophilia, and other complications associated with dysregulation of myeloid responses, especially in the lung [82,83,84]. Epithelial damage is the initial event and hallmark of acute lung injury that initiates a cascade of processes that lead to diffuse lung parenchymal damage [85,86]. We contemplated a scenario for COVID-19 lung immunopathology in which focal airway inflammation produces an elevation of pro-inflammatory mediators. These mediators activate alveolar macrophages and neutrophils, which release ROS and MMPs, producing additional lung tissue damage. The functional significance of the interactions between MMPs and their immunological substrates in the lung is a novel concept that is currently being explored. In particular, we suggested that excessive cleavage of TREM-1 by MMP-8 could contribute to immunosuppression, as demonstrated during other severe infections [87]. Additionally, the action of MMP-2 on sHLA-G release could induce immune impairment and exhaustion [88]. Furthermore, HLA-G expression induced by SARS-CoV2 infection may be associated with increased morbidity and mortality and, as described, predict a worse outcome [89].

This study had undergone several limitations. First, the patients were not monitored for the level of MMPs from admission until recovery. Second, the profile of pro- and anti-inflammatory cytokines was not determined. Third, viral load was not taken into account in the analysis. Fourth, patients with asymptomatic, mild, and moderate COVID-19 were not investigated. Fifth, the small sample size of COVID-19 patients and non-COVID-19 critical controls was another important limitation and may reflect in the statistical significance; and finally, the lack of healthy non-COVID-19 individuals (health control), which could have allowed us to identify a baseline of clinical and inflammatory variables.

## 5. Conclusions

Uncontrolled protease activity and improper expression of several MMPs were correlated to lung disease in severe COVID-19. Although considered plasma prognostic biomarkers, the MMP-2 and MMP-8 pathways in the lung could become the target of specific therapies, including those proposed to diminish cell infiltration, viral immunosuppression response, oxidative stress, and tissue damage during COVID-19. Conversely, MMPs are emerging as an important component of COVID-19 immunopathogenesis.

## Figures and Tables

**Figure 1 biomolecules-12-00604-f001:**
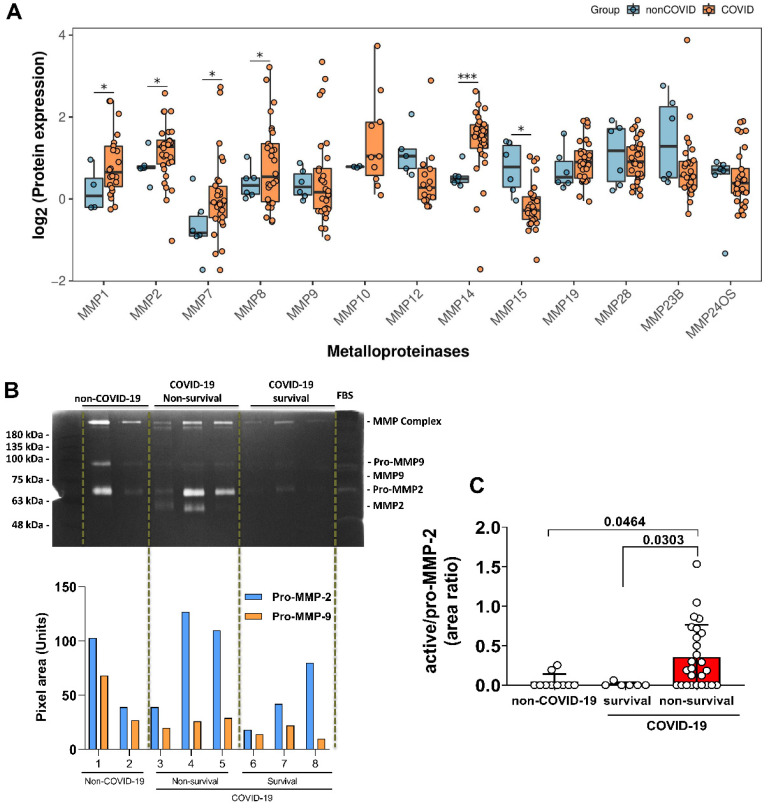
The expression of matrix metalloproteinase and active-MMP-2 form increased in COVID-19 lung. (**A**) Protein expression data obtained from literature [37] was reanalyzed aiming to evaluate MMPs expression changes. Significant changes are indicated by nominal *p*-values *(** *p* < 0.05 and *** *p* < 0.001). Only MMP-14 presented significant difference with adjusted *p*-value equal to 0.000508. (**B**) Expression and different molecular weight forms of MMP-2, cleavage of pro-MMP-2 to active MMP-2, and MMP-9 analyzed by gelatin zymography. Representative gel from non-COVID-19 (*n* = 13), survival (*n* = 6) and non-survival (*n* = 33) COVID-19 patients is shown. Positions of pro-MMP-9, pro-MMP-2 and active MMP-2 with sizes in kDa are indicated. (**C**) Quantification of active-MMP-2 represented by area ratio to pro-MMP-2. Statistical analyzes were performed using the Kruskal-Wallis multiple comparison test, followed by the Dunns post-test to compare pairs. Data are expressed as median with 95% confidence intervals. Statistical differences between groups are considered by *p* < 0.05 and represented directly in the graphic figure.

**Figure 2 biomolecules-12-00604-f002:**
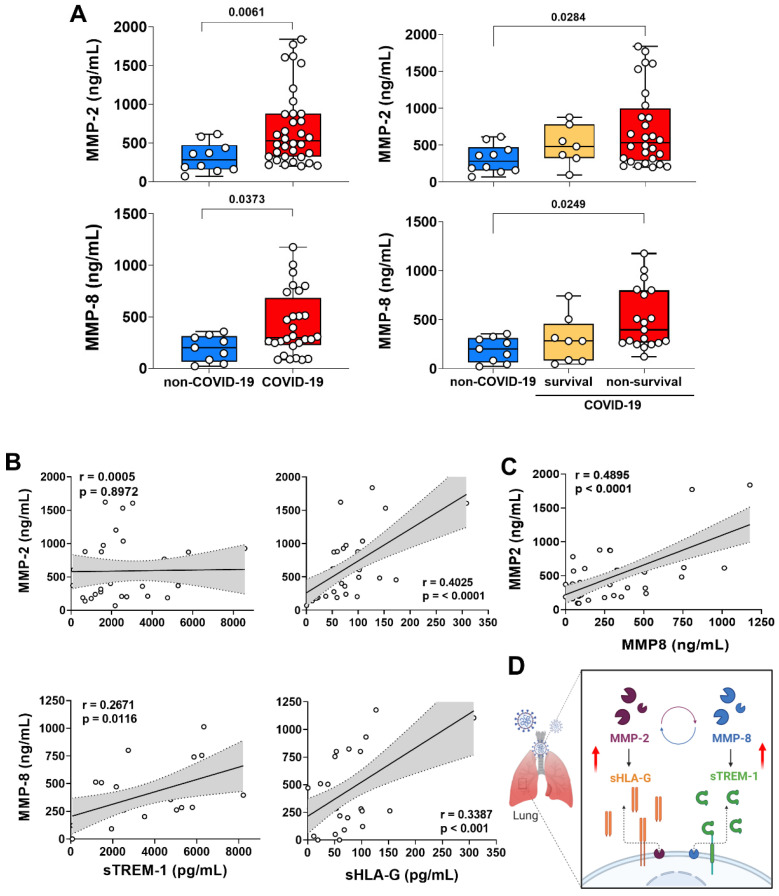
Increased levels of MMP-2 and MMP-8 in TAF samples correlated with non-survival COVID-19 and shed of sHLA-G and sTREM-1. (**A**) Quantification of MMP-2 and MMP-8 in TAF samples from non-COVID-19 (*n* = 13) and COVID-19 patients (survival *n* = 6, non-survival *n* = 33). Statistical analyzes were performed using the Kruskal–Wallis multiple comparison test, followed by the Dunns post-test to compare pairs. Data are expressed as median with 95% confidence intervals. Statistical differences between groups are considered by *p* < 0.05 and represented directly in the graphic figure. (**B**) Correlations between MMP-2 and MMP-8 levels and soluble immune factors sHLA-G and sTREM-1 in total TAF samples from non-COVID-19 and COVID-19 patients at hospital admission. (**C**) Correlations between MMP-2 and MMP-8 levels in total TAF samples from non-COVID-19 and COVID-19 patients at hospital admission. Spearman correlation analysis, *r* and *p* value indicated in each panel. (**D**) Schematic representation of the MMP-2 and MMP-8 positive looping inducing the release of sHLA-G and sTREM-1 in lung from severe COVID-19. (Created with BioRender.com, Agreement number: IZ23RHRAES).

**Figure 3 biomolecules-12-00604-f003:**
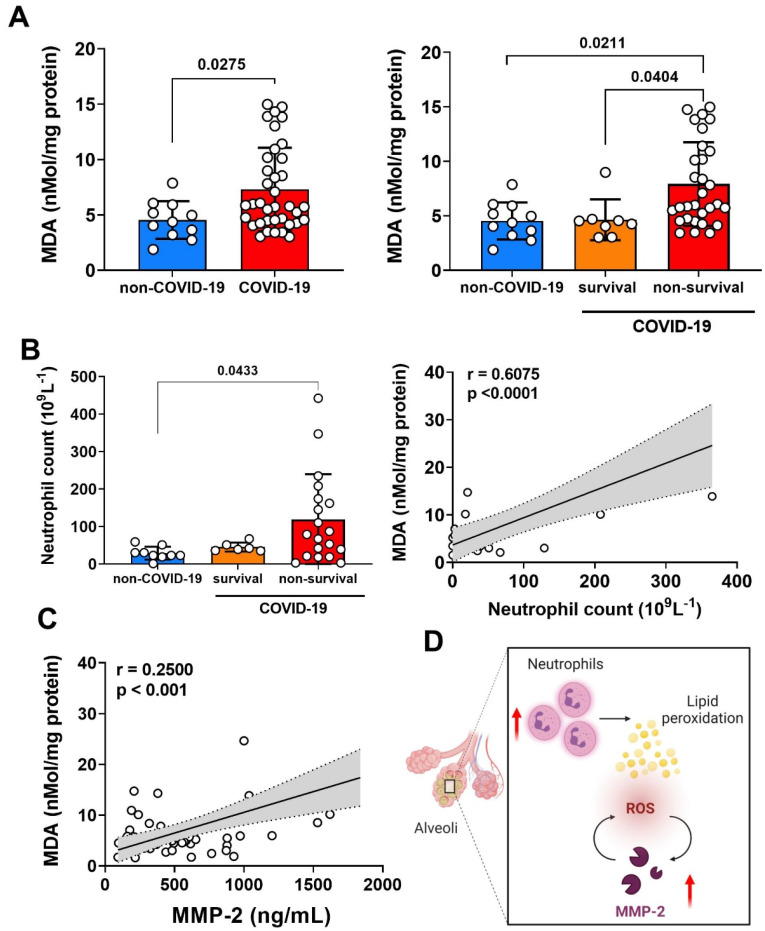
Oxidative stress and neutrophil infiltration into the lung of patients with severe COVID-19. (**A**) Representative lipid peroxidation levels by concentrations of thiobarbituric acid-reactive species expressed in terms of MDA in TAF samples from non-COVID-19 (*n* = 13) and COVID-19 patients (survival *n* = 6, non-survival *n* = 33). (**B**) Absolute neutrophil counts in TAF samples from non-COVID-19 and COVID-19 patients (survival, non-survival), and correlation between the quantification of MDA and the number of neutrophils in total TAF samples from non-COVID-19 and COVID-19 patients at hospital admission. (**C**) Correlation between MDA quantification and MMP-2 levels in total TAF samples from non-COVID-19 and COVID-19 patients at hospital admission. Spearman correlation analysis, *r* and *p* value indicated in each panel. Statistical analyzes were performed using the Kruskal–Wallis multiple comparison test, followed by the Dunns post-test to compare pairs. Data are expressed as median with 95% confidence intervals. Statistical differences between groups are considered by *p* < 0.05 and represented direct in the graphic figure. (**D**) Schematic representation of the lung neutrophil infiltration and ROS production trigger a positive looping of MMP-2 and ROS production in lung from severe COVID-19. (Created with BioRender.com, Agreement number: CG23RHR5O1).

**Table 1 biomolecules-12-00604-t001:** Participants clinical and demographic data enrolled in this study.

Baseline Variable	Non-COVID-19 N = 13	COVID-19 N = 39	COVID-19 Survival N = 6	COVID-19 Non-Survival N = 33	*p* Value
**Demographic characteristics** (**median ± SD**)					
Age	61 ± 17.8	66 ± 16.0	61 ± 24.1	66 ± 14.3	0.5200 ^a^ 0.5488 ^b^
BMI (kg/m^2^)	26.8 ± 6.5	29.4 ± 7.0	27.4 ± 6.7	29.5 ± 7.1	0.1528 ^a^ 0.3915 ^b^
**Sex, No.** (**%**)					
Male	4 (30.8)	21 (55.2)	1 (16.7)	17 (51.5)	-
Female	9 (69.2)	18 (44.7)	5 (83.3)	16 (48.5)	-
**Comorbidities or coexisting disorders, No.** (**%**)					
Hypertension	3 (23.1)	23(59)	3 (50)	20 (60.6)	**0.0250 ^a^** 0.6271 ^b^
Dyslipidemia	-	2 (5.1)	-	2 (6.0)	-
Diabetes *mellitus*	2 (15.4)	15 (38.4)	1 (16.7)	14 (42.4)	0.1245 ^a^ 0.2329 ^b^
Obesity	3 (23.1)	17 (43.6)	1 (16.7)	16 (48.5)	0.1880 ^a^ 0.1482 ^b^
Neurological Disease	2 (15.4)	3 (7.7)	-	3 (9.1)	0.4152 ^a^
Respiratory Disorders	1 (16.7)	5 (12.8)	-	5 (21.7)	0.6162 ^a^
**Presenting symptoms, No.** (**%**)					
Dyspnea	4 (30.7)	27 (69.2)	4 (66.7)	23 (69.7)	**0.0144****^a^** 0.8824 ^b^
Fever	-	12 (30.8)	2 (33.3)	10 (30.3)	0.8824 ^b^
Myalgia	-	17 (43.6)	1 (16.7)	16 (48.5)	**0.0037****^a^** 0.1482 ^b^
Diarrhea	-	7 (18)	-	7 (21.2)	-
Cough	-	13 (33.3)	4 (66.7)	9 (27.3)	0.0597 ^b^
Anosmia	-	4 (10.3)	1 (16.7)	3 (9.1)	0.5737 ^b^
Dysgeusia	-	3 (7.7)	-	3 (9.1)	-
**Laboratory findings** (**median ± SD**)					
Erythrocytes × 10^9^/L	2.7 ± 0.9	3.7 ± 0.8	3.0 ± 0.7	3.9 ± 0.7	**0.0037 ^a^** **0.0136 ^b^**
Hemoglobin (g/dL)	8.7 ± 2.9	10.7 ± 2.2	9.4 ± 1.3	10.9 ± 1.9	**0.0090 ^a^** **0.0115 ^b^**
Leukocytes × 10^9^/L	11.5 ± 15.4	15.1 ± 16.7	11.2 ± 4.2	16.2 ± 18.0	0.5318 ^a^ 0.1850 ^b^
Neutrophils × 10^9^/L	8.7 ± 11.1	12.8 ± 5.5	7.6 ± 4.1	13.4 ± 5.5	0.8028 ^a^ 0.1327 ^b^
Lymphocytes × 10^9^/L	1.1 ± 1.1	0.9 ± 0.5	1.4 ± 0.8	0.9 ± 0.5	0.5117 ^a^ 0.5820 ^b^
NLR	7.0 ± 21.1	12.3 ± 10.5	5.0 ± 10.0	12.8 ± 10.6	0.1879 ^a^ 0.3311 ^b^
Monocytes × 10^9^/L	0.48 ± 0.4	0.52 ± 0.5	0.6 ± 0.4	0.5 ± 0.5	0.7854 ^a^ 0.6482 ^b^
Platelets × 10^9^/L	250 ± 165.8	229 ± 9.15	286 ± 80.7	228 ± 92.3	0.3112 ^a^ 0.6693 ^b^
**Hospital support, No.** (**%**)					
Intensive care unit (ICU)	13 (100)	39 (100)	6 (100)	33 (100)	-
**Hospitalization data, No.** (**median ± SD**)					
Hospitalization days	23 ± 19.6	6.5 ± 2.6	23.5 ± 8.2	17 ± 8.4	**<0.0001 ^a^** **0.0374 ^b^**
**Respiratory support received**					
Invasive mechanical ventilation (%)	13 (100)	39 (100)	6 (100)	33 (100)	-
Oxygen Saturation (median ± SD)	94 ± 6.3	91 ± 10.7	91.5 ± 4.2	91 ± 11.3	**0.0125 ^a^** 0.3240 ^b^
PaO_2_/FiO_2_ ratio (median ± SD)	156.3 ± 58.3	140.3 ± 94.8	106.4 ± 165	140.3 ± 76.4	0.9335 ^a^ 0.9245 ^b^
**Denouement, No** (**%**)					
Survival	5 (38.5)	6 (15.4)	-	-	-
Non-survival	8 (61.5)	33 (84.6)	-	-	-
**Viral charge**					
∆CT	-	57.8 ± 3402	30.2 ± 3899	52.6 ± 10,455	0.3706 ^b^

^a^ Comparisons between non-COVID-19 and COVID-19 patients; ^b^ COVID-19 survival versus COVID-19 non-survival patients. Patient data were compared using the *chi-square* test, or Fisher’s exact test for categorical variables and one-way analysis of variance (ANOVA) Mann-Whitney; nonparametric *t*-test was used for continuous variables. *p* < 0.05 was considered statistically significant. Abbreviations: Standard deviation (SD); percentage (%).

## Data Availability

All data generated or analyzed during this study are included in this published article or Appendix A. Data sharing is not applicable to this article.

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
