# Peer review of "Matrix Metalloproteinases on Severe COVID-19 Lung Disease Pathogenesis: Cooperative Actions of MMP-8/MMP-2 Axis on Immune Response through HLA-G Shedding and Oxidative Stress"

_biomolecules, 2022, doi:10.3390/biom12050604_

Round 1

Reviewer 1 Report

The paper entitled “Matrix Metalloproteinases on Severe COVID-19 Lung Pathogenesis: Cooperative Actions of MMP-8/MMP-2 Axis on Immune Response Through HLA-G Shedding and Oxidative Stress” is well structured and presented. The main topics of the paper are well introduced and the results and discussion are well organized.

In line 289, correct the percentage of death outcome for critical control individuals (61.5%) “The percentage of death outcome was 84.6% for COVID-19 patients and 38.5% for critical control individuals (Table 1).”

Even considering the various limitations of the study, duly presented by the authors, I consider that it is an innovative study, very relevant and that adds knowledge about SARS-CoV-2 infection, with a specific focus on metalloproteinases.

Author Response

The paper entitled “Matrix Metalloproteinases on Severe COVID-19 Lung Pathogenesis: Cooperative Actions of MMP-8/MMP-2 Axis on Immune Response Through HLA-G Shedding and Oxidative Stress” is well structured and presented. The main topics of the paper are well introduced and the results and discussion are well organized.

Answer: The authors thank Reviewer #1 for the positive evaluation of our manuscript.

  1. In line 289, correct the percentage of death outcome for critical control individuals (61.5%) “The percentage of death outcome was 84.6% for COVID-19 patients and 38.5% for critical control individuals (Table 1).”

 Answer: We have revised this point accordingly.

Reviewer 2 Report

  • A brief summary 

The present article titled “Matrix Metalloproteinases on Severe COVID-19 Lung Pathogenesis: Cooperative Actions of MMP-8/MMP-2 Axis on Immune Response Through HLA-G Shedding and Oxidative Stress” is focused on the relationship between MMPs and hyper inflammation of lung tissue caused by coronavirus disease 2019 and the underlying mechanisms.

 Comments 

Introduction

- The introduction is written with a deep knowledge of the studied processes and clearly sets out the purpose of this article by emphasizing the importance of revealing the mechanisms underlying pulmonary inflammation in COVID-19 patients in order to create a new diagnostic biomarker.

Materials and Methods

- This part is written in detail, giving all the details of the methods used.- There is a lack of patient consent to participate in the research

Results

-All figures are presented in very high quality and the data are interpreted appropriately and consistently throughout the manuscript.- The statistical analysis was chosen very precisely 

Conclusion

The conclusion is consistent with the evidence and arguments presented in the article.

As overall the presented article is written very well and the result and discussion are excellently argued with the presented figures and schemes of high quality. Thus, the article will identify and fill the gap in knowledge about the mechanism of action of COVID-19 related to the occurrence of pneumonia, oxidative stress, and MMPs.

Author Response

The present article titled “Matrix Metalloproteinases on Severe COVID-19 Lung Pathogenesis: Cooperative Actions of MMP-8/MMP-2 Axis on Immune Response Through HLA-G Shedding and Oxidative Stress” is focused on the relationship between MMPs and hyper inflammation of lung tissue caused by coronavirus disease 2019 and the underlying mechanisms.

 The introduction is written with a deep knowledge of the studied processes and clearly sets out the purpose of this article by emphasizing the importance of revealing the mechanisms underlying pulmonary inflammation in COVID-19 patients in order to create a new diagnostic biomarker.

Answer: The authors thank reviewer #2 for the positive evaluation of our manuscript

This part is written in detail, giving all the details of the methods used. There is a lack of patient consent to participate in the research.

Answer: We have revised this point accordingly on “2.1 Ethical approval”.

All figures are presented in very high quality and the data are interpreted appropriately and consistently throughout the manuscript. The statistical analysis was chosen very precisely. The conclusion is consistent with the evidence and arguments presented in the article.

 As overall the presented article is written very well and the result and discussion are excellently argued with the presented figures and schemes of high quality. Thus, the article will identify and fill the gap in knowledge about the mechanism of action of COVID-19 related to the occurrence of pneumonia, oxidative stress, and MMPs.

 Answer: The authors thank reviewer #2 for the excellent endorsement.

Reviewer 3 Report

The study/hypothesis generated is interesting. However, the main limitation of the study is very limited no. of samples. The authors need to explore a large number of samples to draw any suitable conclusion. The data presented here is very preliminary. The statistical power is not sufficient enough to draw such conclusions.

Author Response

The study/hypothesis generated is interesting. However, the main limitation of the study is very limited no. of samples. The authors need to explore a large number of samples to draw any suitable conclusion. The data presented here is very preliminary. The statistical power is not sufficient enough to draw such conclusions.

Answer: We agree with the reviewer #3. However, during the pandemic COVID-19, the approach to these severe COVID-19 patients required some biosafety care, and high esteem to the relatives, who endorsed the research ethics consent term. Essentially, among non-COVID-19 patients hospitalized in ICU this access were greater limitations. Thus, those conditions reduced the number of volunteers for this research. The same circumstances are observed in others works with the topic of pulmonary COVID-19 and which have a smaller sample number than we showed in our work (doi: 10.3390/v13060957, doi.org/10.1007/s13238-020-00752-4 and DOI: 10.1080/22221751.2020.1747363). However, we declare the limitations of our study in the last paragraph of the discussion section.  

Reviewer 4 Report

This study examines the involvement of MMPs in the development of severe lung damage associated with COVID-19 infection. While the topic is interesting, I would have serious concerns about the study design, etc.

1.  As a control group, the authors adopted intubated patients who were COVID-19 negative. The primary diseases of the control group are not clearly stated, so they are expected to be a miscellaneous group of diseases. I wonder if it is correct to set them as the control group. Another concern is that the number of control groups is very small (n=13). The setting of the control groups is a critical point that will have a significant impact on the results and conclusions of this study. The authors should perform a fundamental review of the study design.

2. In Table 1, the Control and COVID-19 groups are separately shown, but later in the paper the analysis is divided into the Control group and the COVID-19 survival and non-survival groups. The data in Table 1 should also be divided into the Control group, the COVID-19 survival group, and the COVID-19 non-survival group.

3.  The number of neutrophils in Figure 3B should be listed separately for the COVID-19 survival and non-survival groups.

Author Response

This study examines the involvement of MMPs in the development of severe lung damage associated with COVID-19 infection. While the topic is interesting, I would have serious concerns about the study design, etc.

These are my comments:

  1. As a control group, the authors adopted intubated patients who were COVID-19 negative. The primary diseases of the control group are not clearly stated, so they are expected to be a miscellaneous group of diseases.

Answer: We have revised this point accordingly and the underlying diseases of this group are described in “Supplementary Table 1”.

[…] I wonder if it is correct to set them as the control group

Answer: In fact, our research group suggests this classification for SARS-CoV-2 negative individuals, but it does not exclude pre-existing diseases. Therefore, we have revised this point and used the term “non-COVID-19” only, as mentioned in the graphs and Text.

[…] Another concern is that the number of control groups is very small (n=13).

Answer: We intended to have a negative control group paired with the group of positive patients for SARS-CoV-2, but in the context of the pandemic and sanitary recommendations of isolation and the overcapacity in the intensive care unit (UCI), made it difficult to recruit individuals negative for SARS-CoV-2. These circumstances are also observed in papers with the topic of COVID-19 and which have a smaller sample number than ours. Ex: [doi: 10.3390/v13060957] [doi.org/10.1007/s13238-020-00752-4] [ DOI: 10.1080/22221751.2020.1747363].

The setting of the control groups is a critical point that will have a significant impact on the results and conclusions of this study. The authors should perform a fundamental review of the study design.

Answer: This is a really interesting point. However, we argument some pathologic events that occur only in COVID-19 patients group and our statistical analysis strongly support our data and suggestions.

  1. In Table 1, the Control and COVID-19 groups are separately shown, but later in the paper the analysis is divided into the Control group and the COVID-19 survival and non-survival groups. The data in Table 1 should also be divided into the Control group, the COVID-19 survival group, and the COVID-19 non-survival group.

Answer: We have revised this point accordingly.

  1. The number of neutrophils in Figure 3B should be listed separately for the COVID-19 survival and non-survival groups.

Answer: We thank the reviewer for this suggestion. We have added a graph in the supplemental material (Supplementary Figure 4) demonstrated the recruitment of total cells to the COVID-19 lung, as well as, we modifying the Figure 3B, showing the number of neutrophils for both surviving and non-surviving COVID-19 patients.

Round 2

Reviewer 3 Report

Accepted

Reviewer 4 Report

The authors have revised their original manuscript partly according to the reviewers’ comments. In some points, the authors decided to keep their contents unchanged, however, their rebuttal seems almost reasonable. I would think that this revised manuscript is better organized and suitable for publication.